# Polyacrylic Acid Functionalized Biomass-Derived Carbon Skeleton with Highly Porous Hierarchical Structures for Efficient Solid-Phase Microextraction of Volatile Halogenated Hydrocarbons

**DOI:** 10.3390/nano12244376

**Published:** 2022-12-08

**Authors:** Anying Long, Hailin Liu, Shengrui Xu, Suling Feng, Qin Shuai, Shenghong Hu

**Affiliations:** 1State Key Laboratory of Biogeology and Environmental Geology, School of Earth Sciences, China University of Geosciences, Wuhan 430074, China; 2113 Geological Brigade, Bureau of Geology and Mineral Exploration and Development Guizhou Province, Liupanshui 553000, China; 3Key Laboratory of Green Chemical Media and Reactions, Ministry of Education, Collaborative Innovation Center of Henan Province for Green Manufacturing of Fine Chemicals, School of Chemistry and Chemical Engineering, Henan Normal University, Xinxiang 453007, China; 4Faculty of Materials Science and Chemistry, China University of Geosciences, Wuhan 430074, China

**Keywords:** biomass-derived carbon skeleton, hierarchical structure, polyacrylic acid, solid-phase microextraction, volatile halogenated hydrocarbons

## Abstract

In this study, polyacrylic acid functionalized N-doped porous carbon derived from shaddock peels (PAA/N-SPCs) was fabricated and used as a solid-phase microextraction (SPME) coating for capturing and determining volatile halogenated hydrocarbons (VHCs) from water. Characterizations results demonstrated that the PAA/N-SPCs presented a highly meso/macro-porous hierarchical structure consisting of a carbon skeleton. The introduction of PAA promoted the formation of polar chemical groups on the carbon skeleton. Consequently, large specific surface area, highly hierarchical structures, and abundant chemical groups endowed the PAA/N-SPCs, which exhibited superior SPME capacities for VHCs in comparison to pristine N-SPCs and commercial SPME coatings. Under the optimum extraction conditions, the proposed analytical method presented wide linearity in the concentration range of 0.5–50 ng mL^−1^, excellent reproducibility with relative standard deviations of 5.8%–7.2%, and low limits of detection varying from 0.0005 to 0.0086 ng mL^−1^. Finally, the proposed method was applied to analyze VHCs from real water samples and observed satisfactory recoveries ranging from 75% to 116%. This study proposed a novel functionalized porous carbon skeleton as SPME coating for analyzing pollutants from environmental samples.

## 1. Introduction

Volatile halogenated hydrocarbons (VHCs) such as dichloromethane, trichloromethane, tetrachloromethane, trichloroethylene, tribromethane, etc., have been widely used in industrial applications such as disinfectants, chemical intermediates, and organic solvents [1,2]. However, the waste VHCs would migrate to water, soil, and the atmosphere, thereby resulting in serious harmful issues to human when ingested due to its high toxicity and difficulty of degradation [3,4]. VHCs in water have been listed as priority pollutants by both Chinese and American governments. Therefore, the development of a sensitive and effective analytical method for the monitoring of concentration levels of VHCs in water is essential.

In general, the determination of volatile organic compounds from aqueous samples is implemented by gas chromatography (GC) or gas chromatography–mass spectrometry (GC–MS) [5]. However, sample pretreatments are of the essence prior to detection by instruments [6,7,8]. Among various sample pretreatment techniques, solid-phase microextraction (SPME) exhibited remarkable advantages in the determination of organic compounds from aqueous samples owing to its solvent-free sampling, straightforward operation, and the integration of sampling, extraction, and injection into a single step [9,10,11,12]. In view of these superiorities, SPME has been widely used in environmental analysis [13,14,15], food analysis [16,17,18], pharmaceutical analysis [19,20,21], and biological analysis [22,23,24,25]. The performance of SPME was related to the coating materials as the extraction was based on the equilibrium partitioning of analytes between the sample matrix and fiber coating [26,27,28]. To date, various coating materials have been extensively investigated to improve the extraction efficiencies of SPME, such as activated carbon, metal oxides, graphene, metal–organic frameworks, covalent organic frameworks, ionic liquids, polymers, and other composites [26,29,30,31,32].

Among the developed materials, activated carbon presented unique advantages as a SPME coating for efficiently capturing organic compounds from environmental samples, owing to its high surface area, hydrophobic interaction, low cost, and ease to obtain from agricultural waste [33,34,35,36]. For instance, Ji et al. [37] fabricated a type of nitrogen-doped porous carbon derived from marine algae with a large specific surface area and wide pore size distribution, and used it as an SPME coating for the determination of chlorobenzenes from water. Results showed that the as-prepared SPME fiber exhibited outstanding extraction capacities and the proposed method presented ultra-sensitivity for analyzing chlorobenzenes. Yin et al. [38] proposed a facile method for fabricating SPME coating based on peanut shell-derived biochar, and confirmed its excellent performances for extracting polycyclic aromatic hydrocarbons from water samples. The abovementioned studies demonstrated the effectiveness of biomass-derived carbon as an SPME coating, owing to its high surface area. However, the biomass-derived carbon had undergone a treatment of high temperature, which led to the diminution of chemical groups during the pyrolysis process, thereby resulting in insufficient extraction of slightly polar compounds [39]. Therefore, the decoration of the activated carbon surface with polar groups is an effective means to promote the extraction capacities of VHCs.

Polyacrylic acid (PAA) is a type of eco-friendly polymer with carboxyl (C=O) groups on every two carbon atoms of its main chain. Abundant polar chemical groups endow PAA to be of great potential for adsorbing compounds with polar groups by hydrogen-bond interaction and polar binding [40,41,42,43]. Therefore, we proposed PAA as the donor of polar groups to modify the biomass-derived porous carbon for SPME of VHCs. In addition, discarded shaddock peels are rich in lignin and cellulose, and possess abundant intrinsic pores, thereby becoming an inspiring raw material for the fabrication of biomass-derived porous carbon.

In this study, the nitrogen-doped porous carbons (N-SPCs) were prepared by pyrolysis of discarded shaddock peels at 800 °C under nitrogen protection. Before the pyrolysis process, the shaddock peels were treated by sodium hydroxide and urea solution, where sodium hydroxide acted as activation reagent during the pyrolysis process for the formation of a highly porous structure, and urea was used as the donor of nitrogen. Subsequently, the N-SPCs was modified by PAA—which was labeled as PAA/N-SPCs—and used as SPME coating for determination of VHCs from water coupled with GC–MS (Figure 1). Notably, the as-prepared PAA/N-SPCs exhibited superior extraction capacities for VHCs compared with pristine N-SPCs and commercial SPME coatings owing to its high surface area and abundant chemical groups. Finally, the proposed analytical method was successfully applied for the determination of VHCs from real water samples. To our best knowledge, the as-prepared PAA/N-SPCs was first used as an SPME coating material and applied for the capturing and analyzing of VHCs from water.

## 2. Experimental Section

### 2.1. Reagents, Materials, and Instruments

The VHCs standards, including trichloromethane (CHCl_3_), tetrachloromethane (CCl_4_), trichlorethylene (C_2_HCl_3_), tetrachloroethylene (C_2_Cl_4_), and tribromethane (CHBr_3_), were provided by the Beijing North Weiye Institute of Measuring and Testing Technology (Beijing, China). Urea, sodium hydroxide, and sodium chloride were purchased from Damao Chemical Reagent Factory (Tianjin, China). PAA was obtained from Shanghai Aladdin Biochemical Technology Co., Ltd. (Shanghai, China). Stainless steel wire with a diameter of 100 μm was obtained from Shenzhen Hubei Baofeng Industrial Co., Ltd. (Shenzhen, China). Sylard184 silicone elastomer was purchased from Dow Silicones Corporation (Seneffe, Belgium). Commercial SPME coatings, including PDMS, PDMS/DVB, and DVB/CAR/PDMS, were provided by ANPEL Laboratory Technologies Inc. (Shanghai, China). Shaddock peels were collected from a local market located in Xinxiang, China.

The details of instruments used for characterizations of as-prepared materials including scanning electron microscope (SEM), X-ray diffraction spectrometer (XRD), Fourier-transform infrared spectrometer (FTIR), X-ray photoelectron spectroscopy (XPS), and nitrogen adsorption/desorption apparatus are described in Appendix A. The detection of VHCs was performed by GC–MS (Agilent 7890B–7000D, Santa Clara, CA, USA).

### 2.2. Preparation of PAA/N-SPCs

As displayed in Figure 1, the N-SPCs were prepared by pyrolysis of modified shaddock peels under nitrogen atmosphere. First, the outer yellow layers of shaddock peels were removed to obtain the homogeneous precursors. After that, the shaddock peels were cut into small fragments and dried in a freeze dryer. Then, the shaddock peel fragments were ground into powder. Afterwards, 10 g of shaddock peels powder, 3 g of urea, and 10 g of sodium hydroxide were dispersed in 50-mL water and thoroughly mixed by stirring. After drying in an oven, the modified shaddock peel powder was placed in a horizontal quartz tube furnace (BTF-1200CC-S, Anhui BEQ Equipment Technology Co., Ltd., Hefei, China) for thermal treatment at 800 °C for 2 h with a heating rate of 5 °C/min under nitrogen protection. The obtained powder was washed by hydrochloric acid solution (0.1 M) and deionized water until neutrality. The N-SPCs was then observed after drying in an oven at 105 °C. Then, 1 g of N-SPCs powder was added into PAA aqueous solution for stirring 24 h at room temperature. Finally, the PAA/N-SPCs-x was obtained after being washed by deionized water and ethanol three times, respectively, and dried in an oven, where x represented the mass percentage content of PAA in aqueous solution.

### 2.3. Fabrication of PAA/N-SPCs Coated SPME Fiber

The fabrication procedures were carried out according to our previous studies [13,23,27,44]. Typically, the cleaned stainless-steel wire (3–4 cm in length) was immersed into silicone sealant solution with PDMS polymer and curing agent of 10:1 in mass ratio. Then, the stainless-steel wire was immersed into PAA/N-SPCs powder to form a uniform coating after curing at 120 °C in an oven. Finally, the coated stainless-steel wire was assembled onto an empty SPME needle. Prior to being used for extraction, the as-prepared SPME fiber was aged in the GC–MS injector for 20 min at 250 °C in order to remove the potential interfering compounds.

### 2.4. SPME Procedures and GC–MS Analysis

The SPME process was operated with headspace (HS) mode in a 20-mL commercial vial. First, 10 mL of aqueous solution was added into sample vial, and then the as-prepared SPME fiber was injected into the vial headspace for extraction. During the extraction process, the temperatures were set at 30–60 °C; the extraction time was kept within 20–60 min; the solution acidity was adjusted by hydrochloric acid (0.1 M) and sodium hydroxide solution (0.1 M) with pH values ranging from 3–9; the ionic strength of the solution was controlled by sodium chloride with contents varying from 0 to 20%. After extraction, the SPME fiber was immediately inserted into the GC–MS injector for desorption and analysis. The operating parameters of GC–MS and characteristic ions for analyzing VHCs were detailed in Appendix A.

### 2.5. Collection and Analysis of Real Water Samples

The real water samples were collected from campus tap water (1#) located in Xinxiang and unknown lake water (2#, 3#) located in Anshun, China. The collected water samples were sealed by Parafilm to avoid compounds loss and stored in a refrigerator at 4 °C before analysis by the proposed method.

## 3. Results and Discussion

### 3.1. Characterizations of PAA/N-SPCs

The micro morphologies of both N-SPCs and PAA/N-SPCs were investigated by SEM. As shown in Figure 1a, N-SPCs exhibit a highly porous hierarchical structure assembled by carbon skeletons under the activation of sodium hydroxide. After the modification of PAA, the morphology of PAA/N-SPCs was not changed significantly (Figure 1b). The generated highly porous structure endowed both N-SPCs and PAA/N-SPCs with sufficient specific surface area and accessible contact sites for analytes. It can be seen from optical microscope image (Figure 1c) that a uniform coating of PAA/N-SPCs was formed on the surface of the stainless-steel wire. The thickness of the coating was detected to be 60 μm according to the dimensions before and after coating.

To demonstrate the structural compositions of as-prepared PAA/N-SPCs, XRD analysis was performed (Figure 2a). It can be seen that the XRD patterns of both pristine N-SPCs and PAA/N-SPCs display two peaks at 24.7° and 29.4°, which correspond to the diffraction peaks of amorphous carbon and NaO2, respectively [45]. The XRD results suggest that the as-prepared porous carbon skeletons have not undergone significant structural change before and after the modification. FTIR analysis was carried out to investigate the surface functional groups and the impact of PAA addition on the chemical structure of the N-SPCs (Figure 2b). As shown in the spectrum of N-SPCs, the broad band at 3437 cm^−1^ is due to the O–H stretching, and the peaks at 2927 cm^−1^ and 2852 cm^−1^ correspond to the presence of –CH_2_– bond. The absorption peak at 2355 cm^−1^ belongs to the stretching of C≡N [46]. It can be found that, compared to the characteristic stretching vibrations of C=O located at 1620 cm^−1^ for N-SPCs, the C=O band shifted to a much lower wavelength (1554 cm^−1^) for the PAA/N-SPCs, demonstrating the strong hydrogen bonding between PAA and N-SPCs [47]. Moreover, XPS analysis was employed to further explore the chemical groups of PAA/N-SPCs. The XPS survey spectrum (Figure 2c) confirms the presence of C, O, and N at 283.8, 532.3, and 400.1 eV with atom contents of 78.0%, 19.7%, and 2.3%, respectively. High-resolution spectrum of C 1s (Figure 2d) were deconvoluted into four peaks at binding energies of 284.0, 284.7, 285.5, and 288.7 eV, which were ascribed to C=C, C–C, C–N, and COOH, respectively [48]. The XPS spectrum of O 1s (Figure 2e) was divided into two peaks at 531.7 and 532.4 eV, referring to C–O and O–C=O groups, respectively [49]. N 1s spectrum (Figure 2f) was deconvoluted into two peaks for C–N and N–H at 398.9 and 400.3 eV, respectively [50]. The generated oxygen/nitrogen-containing groups promoted the binding with compounds containing polar groups [39,44].

In general, high surface area and porosity are conducive to provide more active adsorption sites for analytes. Therefore, the specific surface areas and pore size distributions of both N-SPCs and PAA/N-SPCs were evaluated using N_2_ adsorption–desorption and BJH isotherms. As displayed in Figure 3a, the specific surface areas of N-SPCs and PAA/N-SPCs were calculated to be 517.7 and 485.7 m^2^/g, respectively. The average pore diameters of both N-SPCs and PAA/N-SPCs (Figure 3b) were observed with values of 3.6 and 405.2 nm, respectively, which confirmed the formation of both meso- and macro-porous structures. The mesopore and macropore volumes were obtained with values of 0.15 and 0.63 cm^3^/g for N-SPCs, and 0.13 and 0.58 cm^3^/g for PAA/N-SPCs, respectively. Plentiful porosity provided numerous channels for analytes transferring from the outer surface to inside. Although the specific surface area of PAA/N-SPCs was slightly lower than that of N-SPCs, the formed abundant chemical groups on PAA/N-SPCs, which was verified in FTIR and XPS analysis, improved the adsorption capacities for VHCs by polar interaction. Moreover, the thermal stability of as-prepared materials is essential as the desorption of SPME is performed in a GC–MS injector at 250 °C. Herein, the desorption of the as-prepared new SPME fiber in the GC–MS injector at 250 °C was carried out with scan mode (Appendix A). Results showed that no distinct interfering compounds were found, indicating the excellent thermal stability of the as-prepared SPME coating.

### 3.2. SPME Capacities of PAA/N-SPCs Coating for VHCs

The extraction efficiencies of as-prepared SPCs, N-SPCs, and PAA/N-SPCs coatings for five VHCs from water were investigated. As shown in Figure 4a, the extraction efficiencies of PAA/N-SPCs-5% exhibit an evident enhancement for VHCs compared with pristine N-SPCs and SPCs—especially for C_2_Cl_2_, owing to the functionalization of polar chemical groups by PAA. Typically, excess polar chemical groups on the materials would decrease the hydrophobic interaction with analytes. Figure 4b displays the GC–MS chromatograms of VHCs observed by PAA/N-SPCs-5% and commercial SPME coatings. Results demonstrate that the extraction efficiencies of PAA/N-SPCs are much higher than that of commercial SPME coatings including PDMS, PDMS/DVB, and DVB/CAR/PDMS. The outstanding extraction performance of PAA/N-SPCs toward VHCs can be ascribed to two factors: (i) high specific surface area and meso/maro-porous hierarchical structures promoted the adsorption active sites; (ii) polar chemical groups (C–O, O–C=O, –CN) enhanced the interaction with VHCs by hydrogen bond and polar binding; where, the extraction capacity of commercial coatings was mainly attributed to the hydrophobic crosslinking. It is worth noting that although polar binding was formed between the analytes and the coating material, the adsorbed analytes can be easily desorbed in the GC–MS injector at 250 °C.

### 3.3. Optimization of SPME Conditions

To observe the finest extraction efficiencies, the effects of SPME conditions including extraction temperature, extraction time, ionic strength, and pH of solution were studied using a spiked aqueous solution with a VHCs concentration of 10 ng mL^−1^. Figure 5a presents the effect of temperature on the extraction efficiencies. It can be seen that the extraction efficiencies exhibit a decreased trend with the temperature increase from 30 to 60 °C. Although higher temperature can promote the release of analytes to the headspace, the distribution coefficients of analytes on the fiber coating diminish. Further, as a type of highly volatile compound, VHCs can be released to the headspace from water at a low temperature. Therefore, the extraction efficiencies of PAA/N-SPCs decrease gradually in the temperature range of 30–60 °C. SPME with headspace mode is an equilibrium-based process of analytes among the sample matrix, headspace, and fiber coating. The effect of the extraction time ranging from 20 to 60 min on extraction efficiencies was evaluated (Figure 5b). Results imply that the extraction process reaches equilibrium within 40 min. High ionic strength of solution can enhance the analytes’ release from the matrix, resulting in an increase in extraction efficiency. Herein, sodium chloride was used to control the ionic strength of water. As displayed in Figure 5c, sodium chloride in water with a content of 5% improves the extraction efficiency, whereas excess contents of sodium chloride weaken the extraction capacity due to the adhesion of sodium chloride on the coating surface. The effect of the solution pH ranging from 3 to 9 (Figure 5d) presents an unapparent change in extraction efficiencies of VHCs, except for tetrachloroethene, which demonstrates that the PAA/N-SPCs coating can be applied in a wide acidity range. To sum up, the optimum extraction conditions of PAA/N-SPCs coating with a temperature of 30 °C, extraction time of 40 min, sodium chloride content of 5%, and pH of 7 were performed for method validation.

### 3.4. Analytical Method Performance

Under the optimal extraction conditions, the method performances for analyzing VHCs from water were evaluated by means of linearity, limits of detection (LODs), limits of quantitation (LOQs), and relative standard deviations (RSDs). As listed in Table 1, the proposed method presents wide linearity in the VHCs concentration of 0.5–50 ng mL^−1^, with linear coefficient (R^2^) of 0.9879–0.9973. The LODs and LOQs were calculated to be 0.0005–0.0086 ng mL^−1^ and 0.0015–0.029 ng mL^−1^, according to three and 10 times of signal-to-noise, respectively. The reproducibility of method was evaluated by RSDs, with results that the RSDs with five replicates range from 5.8% to 7.2% for analyze VHCs from water samples. Excellent linearity, high sensitivity, and good reproducibility suggest the proposed method has great potential in real water analysis.

### 3.5. Real Water Samples Analysis

Finally, the proposed method based on PAA/N-SPCs SPME fiber coupled with GC–MS was applied to measure VHCs from real water samples. As listed in Table 2, no VHCs were found in campus tap water (1#). Trichloromethane and tribromomethane were detected with concentrations of 8.0 and 8.5 ng mL^−1^ in Sample 2#, respectively; trichloromethane and tetrachloroethene were found with concentration of 7.5 and 12.0 ng mL^−1^ in Sample 3#, respectively, which were collected from an unknown lake. The recoveries were obtained in a range of 75%–116% with spiked concentration of 0.5 ng mL^−1^ for sample 1#, and 5 ng mL^−1^ for the other samples. Satisfactory recoveries confirmed the effectiveness of the proposed method in analyzing VHCs from real water samples.

## 4. Conclusions

In summary, PAA-functionalized N-SPCs with highly meso/macro-porous hierarchical structure, large surface area, and abundant chemical groups, was fabricated and innovatively used as an SPME coating for extracting VHCs from water. The decoration of PAA enhanced the extraction capacities of VHCs compared with N-SPCs pristine, owing to its abundant polar chemical groups that promoted the interaction with polar group-containing compounds by hydrogen bonding and polar binding. In view of these distinct advantages, the proposed analytical method based on PAA/N-SPCs SPME fiber presented good linearity, excellent reproducibility, and high sensitivity for analyzing VHCs from water. This study confirmed the effectiveness of PAA/N-SPCs as an SPME coating for the capture and analysis of trace pollutants from aqueous samples.

## Data Availability

This study presents novel concepts and did not report any data.

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
