# Peer review of "Polyacrylic Acid Functionalized Biomass-Derived Carbon Skeleton with Highly Porous Hierarchical Structures for Efficient Solid-Phase Microextraction of Volatile Halogenated Hydrocarbons"

_nanomaterials, 2022, doi:10.3390/nano12244376_

Round 1

Reviewer 1 Report

The submitted manuscript reports the synthesis of a new coating material for the determination of volatile halogenated hydrocarbons by SPME. The authors prepare a functionalized carbon-based material starting from a citric fruit peal that was impregnated with urea (source of N) and NaOH (chemical activating agent) and further treated at high temperature under nitrogen flow. The obtained solid was washed and impregnated with PAA in the presence of PDMS to enhance the number of surface surface groups. This is an interesting work and the results show the improved performance of the newly developed SPME fiber in comparison with commercial SPME coatings.

There are however some points that need to be improved before considering publication:

1.    Line 89 explain the use of urea as co-reagent

2.    Line 119 explain why only the wight part of the peel is used

3.    Line 121/122: clarify if urea and NaOH was both added to the same 50 mL in order to obtain the mentioned molar concentration or if two independent solution of 50 mL were mixed.

4.    Line 130 percentage in mass or volume?

5.    What is the role of N-doping during the synthesis of SPCs for the overall process? A N-free material should have been prepared and tested against the N-SPC and PAA/N-SCP.

6.    Line 139 clarify if it is PDMS or the precursor of PDMS. If is the last case, authors need to specify at least the name of the compound added.

7.    Lines 142/143 why this pre-treatment? Clarify in the revised version. Did the authors verify the release of thermolabile by-products resultant form the production of the new fiber?

8.    Section S1: information missing in all the techniques. SEM analysis as prepared or covered with a conducting metal? Operating conditions for XRD and FTIR, specify if FTIR was made with KBr pellets or not. N2 adsorption needs to specify degassing conditions (time and temperature) and mass of material analysed. The reference for BET area determination must be added and the authors must explain the criteria for the selection of the p/p0 range used in the BET area calculation.

9.     Lines 163/164 SEM microscopy imagens do not allow to observe mesopores and macropores. The sentence in the original manuscript is not acceptable and must be removed along with the info added to figure 1a) and further conclusions in lines 207-209 and others along the text. SEM scale is at the micrometer range while nanopores have apertures lower than 100 nanometers.

10.  FTIR analysis: band at ~3400 cm-1 is characteristic of O-H and not N-H, also peak at ~2355 cm-1 results from CO2 and not C-N as authors state. Moreover, authors cite ref 46 to support this the attribution of peak at ~2355 cm-1 to C-N but that manuscript has not peak at that wavelength.

11.  Micropore and mesopore volumes should have been quantified. Given the error associated with the determination of BET area volume the value reported must not present decimal cases.

12.  Lines 221-223: it is only true for 5% and C2Cl4.

13.  Line 231 to ascribe to the listed two factors authors must at least report the characteristics of the commercial fibers as presented by the fabricant or in literature.

Author Response

Thanks for your comments and suggestions. They are very helpful for us to improve the work. Please check the attached file for the point-by-point response.

Reviewer 2 Report

1. The parameters of all of the experiments  should be described in details. For example, the conditions for X-ray diffraction analysis - type of radiation, wavelength, 2Ï´ range, step est.
2. The X-ray diffraction method gave us a  diffraction pattern or diffraction data for a given material but not a spectra, because we use a single wavelength (not a spectrum).

Author Response

(The authors gave the same response as above.)

Reviewer 3 Report

Reviewer report

Manuscript ID: nanomaterials-2028398
Type of manuscript: Article
Title: Polyacrylic Acid Functionalized Biomass-derived Carbon Skeleton with
Highly Porous Hierarchical Structures for Efficient Solid-Phase
Microextraction of Volatile Halogenated Hydrocarbons

In this Manuscript, polyacrilic acid functionalized N-doped porous carbon was fabricated from shaddock peels for volatile halogenated hydrocarbons extraction from the water. The manuscript is well-organized; however, it can be accepted for publication after addressing to the following comments: 

Comments

1.       What is the role of urea and NaOH in preparation technology?

2.       Why PAA was chosen as a decorating agent? What is advantage of PAA/N-SPC compared to the activated carbon, which has much higher specific surface area (up to 1800 m2/g)?

3.       Line 167: Large specific surface area… It is large compared to what?

4.       The authors claim, that the decoration of PAA enhanced the extraction capacity, due to its polar groups. This part must be detailed, since it is the key point of this paper. Does it mean, that a chemisorption partially occurs during the process? In this case, the desorption processes must go more hardly due to high bonding energies.  Explanation is needed.

5.       Line 207: The meso-/macro-porous hierarchical structures of the prepared materials…. To show meso-/macro-porous structure, the authors must modify the abscissa to the large range (up to 150 nm), since macro pores starts from 50 nm (Fig. 3b). From Fig. 3b I cannot see macropores distribution. SEM image is not enougth.

Author Response

Thanks for your comments and suggestions. They are very helpful for us to improve the work. The point-by-point response is provided as the attached file.

Round 2

Reviewer 3 Report

I am satisfied with the response provided by the authors and the revision made to the manuscript. The manuscript can be accepted for publication.